# Educational, Emotional, and Social Impact of the Emergency State of COVID-19 on Romanian University Students

**DOI:** 10.3390/ijerph19073990

**Published:** 2022-03-27

**Authors:** Cristina Gavriluță, Costel Marian Dalban, Beatrice Gabriela Ioan

**Affiliations:** 1Department of Sociology and Social Work Department, Alexandru Ioan Cuza University, 700506 Iasi, Romania; cristina_gavriluta@yahoo.fr; 2Department of Legal Medicine, Faculty of Medicine, Grigore T. Popa University of Medicine and Pharmacy, 700115 Iasi, Romania; beatrice.ioan@umfiasi.ro

**Keywords:** COVID-19 pandemic, students, educational dimension, emotional dimension, social dimension

## Abstract

*Background and Objectives*: The COVID-19 pandemic has had a global impact at the social, economic, cultural, and political levels. Education is one of the areas that experienced a sudden change during the COVID-19 pandemic that affected both students and teachers worldwide. Thus, the aim of our research was to analyze the educational, emotional, and social impact of the period of the emergency state (16 March 2020–15 May 2020) imposed by the COVID-19 pandemic on Romanian university students. *Materials and Methods*: We conducted a questionnaire-based survey among Romanian university students at the national level. *Results*: Our study showed that students accepted online education only as a form of compromise in relation to the epidemiological situation. However, they were affected by the diminished contact with the university and their colleagues and the lack of a regular routine. Emotionally, the participants experienced feelings of loneliness, panic, fear, aggressiveness, and intolerance due to the lack of cultural activities, the struggle with the usual routine, and the restriction of communication and movement. Socially, the relationships with friends and university colleagues were affected; many students returned home to their parents, who supported them during the lockdown. As a general finding, our study describes a social category that felt the full effects of isolation during the emergency state but still managed to cope with the situation by mobilizing a number of specific resources: family, intellectual and cultural concerns, and faith. *Conclusions*: The emergency state imposed by the COVID-19 pandemic has been a special experience in the lives of Romanian students. Its dramatism was tempered well by an effective support mechanism provided by social ties, intellectual formation, and a certain religiosity. This has produced good resilience among students, but also in the communities they belong to.

## 1. Introduction

### 1.1. Higher Education during the COVID-19 Pandemic

The COVID-19 pandemic has affected the global population, and it has produced changes at the social, economic, cultural, and political level [1]. Education is one of the fields that experienced a sudden change in the epidemiological context in 2020. Since 25 March 2020, 150 countries have closed schools nationwide, affecting more than 80% of the world’s student population [2]. According to UNESCO, “at least 1.5 billion students and 63 million primary and secondary teachers are affected by the unprecedented interruption caused by the COVID-19 pandemic, with the closure of schools in 191 countries” [3].

The International Association of Universities showed that the teaching process was shifted to almost exclusively online throughout the world. This sudden change determined a number of challenges, such as: access to digital infrastructure, digital skills required by the use of online platforms, and difficulties in fulfilling certain requirements of specific disciplines [4]. Moreover, the lack of adequate expertise in achieving hybrid or online education, unequal access to the online platforms, limited internet access, lack of appropriate tools, underdeveloped skills and competences in online work, inappropriate content for the new way of working, and the lack of direct interaction would all become a real problem for students, but also for teachers. All these shortcomings that have increased the difficulties of adapting to new conditions and inequities are also emphasized in the *Education Report During the Pandemic. Responses to the Endless Crisis of Romanian Education* [5].

A study conducted in China revealed that adapting to the online teaching format is challenging, especially for the courses which have been designed for face-to-face implementation such as physical education courses. The 54 interviewed students stated that they were not familiar with online education, the content of the courses was not adapted, and the difficulties in understanding the information were much greater [6].

Another longitudinal study from China shows important changes in social mentality at the level of students. Questionnaires were administered in five stages, and the social mentality of the students changed significantly from questionnaire 1 to questionnaire 5. The lowest level was found in the replies given to questionnaire 3, and there were significant increases at level 4, with the highest level in questionnaire 5 [7].

In Europe, a study conducted at the University of Debrecen, Hungary found that distance learning was perceived as a challenge both by teachers and students. Simultaneously, online learning was perceived as an opportunity to experience new ways of teaching/assessment. The proportion of respondents with access to the internet was 94% in Hungary [8].

A comparative analysis between Spain (Europe), Chile (Latin America), and Jordan (Asia) showed that socio-economic, technological, and cultural valences influenced the way students understood and adapted to new educational forms [9]. In December 2020, the impact of teaching online on a total of 588 students from the American University of Sharjah in the United Arab Emirates was analyzed. The psychological sufferings of the students were mainly caused by the quality of the courses, the academic performance, and the preparation for future studies [10].

The emotional impact on students was captured at the Faculty of Medical Sciences of the Medical University of Silesia in Katowice, Poland. Here, the students reported a 67.7% increase in stress levels due to the pandemic context. A measure proposed to solve this problematic issue was the intervention of the university in the psychological care of students through telemedicine [11]. With regard to the mental health of students, there are also studies that were conducted before the epidemiological context [12,13,14].

A study conducted in the UK found that students experienced a more difficult relationship with universities, additional financial costs, discrepancies in the existence of digital skills, a heavier workload, and a low level of well-being. To reduce these effects, teachers performed 85% of the synchronous activities both online and offline [15].

### 1.2. Directions of Analysis

There are three dimensions in which the pandemic affected students, consequently generating specific effects:
The educational dimension, which radically changed the forms of teaching, learning, and assessment. A study conducted at the University of Seville in Spain showed that adapting to the online teaching format can also be explained by the students’ perception of the usefulness of products and learning outcomes, with 70% of the participants showing interest in the need for good digital materials in classes. [16]. About 50% of students had limitations in adapting to this new style of learning. Students have accepted these educational measures to reduce the risk of interruption in their studies [17]. Studies also found challenges in moving into the virtual environment, the lack of online assessment tools, or the difficult access to technology [17].The emotional dimension, which generated emotional imbalance or changes in attitudes towards different aspects of life. The IRES research of February 2021 attests to the fact that young people are a vulnerable category in a pandemic context; cumulatively, 39% of young people (18–35 years) experienced anxiety [18]. Emotional dimming was perhaps the most frequently experienced state of mind among students, and its analysis with different indicators is essential in understanding the context of confinement during the COVID-19 pandemic. A study conducted in Lithuania showed that students experienced high levels of anxiety, depression, and suicidal thoughts [19]. Another study which involved students in the Faculty of Engineering in Sibiu, Romania integrated elements related to the emotional aspect and the changes that occurred during the lockdown, and identified high variations in stress, anxiety, and tension [20].The social dimension is characterized by relational, economic, and professional problems and the transition to adulthood, which are catalytic elements in the process of increasing psycho-affective disorders [21]. The pandemic has affected the lives of students on multiple dimensions, and they have had to find appropriate coping mechanisms. For instance, in the Netherlands, a study on PhD students found that in order to adapt to the lockdown, students maintained a healthy lifestyle by increasing physical activities or adopting a healthy diet [22].

Our research targeted Romanian university students. They represent an important social category for at least three reasons. Firstly, they belong to the education sector, which has been deeply affected by the state of emergency and lockdown measures. Secondly, students have high mobility through national/international travel and participation in various Erasmus programs and a desire for affirmation specific to their age and professional interests. Thirdly, students have access to information, knowledge, and analytical skills that go beyond average due to their training. Thus, a study of their perceptions and, implicitly, of the effects of the emergency state on students can provide the perspective of an educated population with a certain vision and with specific expectations. Lifestyle and feelings experienced by young people during the emergency state can influence decisions as well as shape attitudes and behaviors.

The sudden closure of schools in March 2020 and the switch to the online courses in universities combined with the state of emergency restrictions have placed students in an unusual situation that they have never encountered before. Thus, the whole Romanian educational system has had to face a new challenge: digital and online education.

Within this general framework, the aim of our study was to identify Romanian university students’ perception of the emotional, social, and educational impact that the state of emergency had on them. The way they assessed it and the way they reacted to it may be a useful and actual barometer during the state of emergency.

## 2. Materials and Methods

### 2.1. Research Objectives and Questions

#### 2.1.1. Research Objectives

To measure the impact of the quarantine period on the academic activities of the university students;To identify the specific emotional patterns of the students during the period of isolation;To identify the different manifestations of the social relations of students during the isolation period.

#### 2.1.2. Research Questions

How do students assess the changes during isolation (lockdown, quarantine, etc.) in relation to educational activities?How do students feel about the impact of isolation on their lives?What does the social life of students in isolation look like from the relational and socialization perspectives?

### 2.2. Data Collection

We conducted a quantitative research by means of an online, questionnaire-based survey. Using the “Google forms” platform, the questionnaire was distributed to universities across Romania (Appendix A). The questionnaire was structured into three main dimensions of analysis, each of which corresponds to a number of indicators and specific questions (Table 1).

Each research dimension corresponded to a set of questions—semi-open, closed, and scaled, with single or multiple answers. The scaled questions were intended to highlight experiential nuances and their perception of the pandemic as well as its educational, emotional, and social consequences. Some indicators were measured through a Likert scale between 1 and 5, where 1 meant a lack of agreement (small extent) and 5 entailed a high level of agreement/a large extent. We used this type of the scale to measure subjective feelings, states, and attitudes related to the importance of individual wellbeing during the period of isolation (lockdown). Additionally, through these types of subjective data, we tried to observe the level of polarization of the social attitudes related to the pandemic context. Our intention was to illustrate, with this type of questionnaire, personal attitudes and emotions related to a complex context that had been generated by the COVID19 pandemic [23,24,25].

We associated a number of indicators to each dimension. In each of these dimensions, we included indicators that we described and presented in Table 1.

The educational dimension refers to the relations between students and teachers; teaching, learning, and assessment; and the open access to educational resources.The emotional dimension refers to the well-being and emotional state of the students during the period of isolation.The social dimension is related to the natural occurrence of socialization and interaction between individuals. Moreover, this dimension stresses the importance of secondary socialization, social network, and the magnitude of the social interactions.

Data were collected anonymously during the emergency state, between 20 April and 10 May 2020, when Romania was at the peak of imposing restrictive measures to decrease the spread of the SARS-CoV-2 virus. Therefore, the responses provided by the research participants will be unaltered by the passage of time and by a retrospective reflection on the moment, or even by a certain habit that might be formed in the case of a prolonged lockdown. Measuring reactions and attitudes at the very moment of an event gives the collected data more credibility and value in understanding the studied phenomenon. The answers received would bear the living imprint of the moment as they are not influenced by other events, emotions, or experiences.

### 2.3. Data Analysis

The statistical analysis was carried out with the Statistical Package for Social Sciences (IBM, version 20), and for the analysis of the data, we used elements of descriptive statistics such as mean, relative frequency, cumulative relative frequency, minimum and maximum value, and standard deviation.

### 2.4. Study Participants

The study group included students from all university centers in Romania.

We used a theoretical sample of convenience, where the most common selection criterion was based on the accessibility of the participants. The motivation for choosing this methodological approach was influenced by the epidemiological context, which imposed a series of isolation rules, making the access to universities difficult. However, collecting information during the very period of isolation was an advantage in that we could obtain direct information, untainted by the passage of time and other experiences. Moreover, we believe that the large number of respondents coming from all university centers in Romania provides us with a picture of the moment regarding the impact of isolation on a homogeneous social category. Such techniques are frequently used for the study of homogeneous populations such as the target population in our study [26].

The invitation to participate in this study was distributed through student associations from each university center (Appendix B), which helped us to reach all university centers in the country, ensuring a balance of respondents in the socio-humanistic and scientific fields, and to proportionally gain access to the students enrolled in different study programs (bachelor’s, master’s, doctoral).

The study group included 1013 participants. An analysis of the study participants according to age, field of study, living environment, and university program in which they are enrolled indicates a good coverage of these dependent variables (Table 2). The average age of the participants was 21 years, with a minimum of 18 years and a maximum of 56 years. A total of 58.7% of respondents study in the field of science, and 41.3% are enrolled in the socio-humanities field. Out of all respondents, 83.1% are enrolled in a bachelor’s program, 13.8% in a master’s, and 3.1% in a doctoral program. A total of 61.9% of the participants were from rural areas, while 38.1% were from urban areas.

## 3. Results

We structured the presentation of the results according to the three dimensions mentioned above, which allowed us to capture the experiences of the students on multiple levels and in the complexity of their manifestations.

### 3.1. Educational Dimension

Our research shows that most of the students (92.6%) managed to maintain contact with the educational institution and teachers. Online platforms, telephone, e-mail, or social platforms were an important resource for dealing with academia as direct in-person meetings had greatly diminished [27]. Only 7.4% of the participants said they had failed to maintain a direct link or a link via an online platform (Table 3).

The number of those who had difficulty maintaining links with the university is higher in rural areas.

Regarding the format of the classes, Romanian students clearly expressed their opinion in favor of the on-site (classical) courses (57.8%), which conveys that direct and live interaction is needed to ensure greater efficiency in learning and receiving explanations from teachers; 55.1% considered that online learning, teaching, and assessment should only be a crisis solution (Figure 1).

Our research also shows that studying, reading, doing sport, and especially communication with those close to them were activities that most students did not give up during the state of emergency (Figure 2).

Of the participants who did not keep in touch with the university, 66% also stated that their relationship with their university colleagues was affected. Moreover, 52% of those who failed to attend the online courses were affected by the lack of a regular routine; the lack of cultural activities, communication, and meetings with colleagues and friends; and the lack of movement (Figure 3).

### 3.2. Emotional Dimension

The pandemic crisis, especially during the peak of restrictions, has primarily impacted people’s emotional states, perceptions, and behavior. This is also the case for students. Thus, 623 (61.5%) respondents considered this period to be a real challenge for them, and 642 (63.3%) considered it to be a trial, a test that they had to overcome.

A number of variables used in our analysis indicates a greater sensitivity among Romanian students towards the social and cultural components of life as parts of the emotional dimension. In this respect, the following variables were considered: loneliness, news, people with whom they lived, lack of a regular routine, uncertainty, lack of communication, lack of cultural activities, decrease/lack of income, lack of physical interaction with colleagues and friends, and lack of movement.

The importance of these variables was graded by the respondents using a 4-point scale, from 1 (very little) to 4 (very much). We considered the variables that had an average of 2 or more as having a high impact on students. Using a multi-answer question, the number of selections described a strong interest in the socio-cultural component, to which a certain pessimism related to the lack of perspective associated with the moment of crisis was added. The most frequent problem identified by the participants was the lack of physical interaction with colleagues, followed by the lack of movement, lack of cultural activities, lack of ordinary routine, lack of perspective, and loneliness (Table 4).

Therefore, the lack of perspective and major changes in the education system generated a state of uncertainty, concern, and stress.

Students’ reactions during the isolation period and their frequency cover a wide range of expressions of emotional states that they experienced. Of these, boredom (with a very high frequency), irritability, and fear exceeded the average value of manifestations on a scale from 1 (rare) to 5 (very often). The other manifestations (intolerance, aggression, bouts of crying, abandonment, despair, and panic) recorded quite high scores, approaching the mean value of manifestations (Table 5).

It is worth noting that the fear of getting infected led many students to pray to God for their health and that of their loved ones: 32.1% daily, 12.3% weekly, 35.6% sometimes, and 19.9% not at all.

Moreover, the fact that a large number of students (N = 642) perceived the isolation period as a trial of life indicates the presence of a series of temperate mental states, yet some perceived it as a chance to be released from many obligations (N = 444) and as a chance for rediscovery (N = 509) as well. These answers indicate a bright and constructive perspective despite a marked sense of frustration.

The general reaction of the students is subsumed under a moderate attitude of reserved optimism: 664 (65.5%) respondents considered that the danger would not end with the termination of the isolation period, and 303 (29.9%) considered that there would also be a second wave (a fact proven between August and September 2020). All of this rather signals certain expectations about the evolution of the pandemic and what will follow.

Our results show that the experience of isolation encouraged many participants to value what they have, their family, and loved ones more (81.5%), as well as to change their priorities (64.3%).

### 3.3. Social Dimension

Regardless of the effects of the pandemic context, 84.9% of the participants considered that the state of emergency, and implicitly the lockdown, was a good solution to reduce the spread of the virus.

With universities and university campuses closed, 72.2% of students said they lived with their parents during the crisis, 9.3% alone, 6.9% with their spouse, 8.4% with a friend, and 3.3% with a fellow student. Thus, the student way of life has changed, with family becoming a place of protection, safety, and refuge for many: 56.8% of students said that during the isolation period, they were supported by parents, 24.5% by a loved one, and only 9.9% by friends.

On a scale from 1 (not at all) to 4 (very much), students’ relationships with parents, neighbors, colleagues, siblings, relatives, and friends during the crisis were analyzed. Through their answers, the students evaluated the extent to which their relationship with the actors mentioned in Table 6 had been affected.

The least affected were the relations with neighbors, with an average response of 1.41, followed by relations with siblings, parents, and relatives (1.53).

Of the 665 students (representing 65.6% of participants) who considered that their relations with their parents did not suffer during the lockdown, 431 (42.5%) considered that their parents had become the closest to them during those difficult times.

## 4. Discussion

In this research, we aimed to explore the impact of the emergency state imposed due to the COVID-19 pandemic on Romanian students by approaching three major dimensions: educational, emotional, and social.

Although they may represent totally different areas of analysis at first glance, there is a motivation for their selection: the health and protection measures imposed during the emergency state primarily affected the daily life and the traditional activities of students. The educational process underwent major changes, being transposed into a totally new and different online format. Universities were closed and most students returned home, thus creating an almost total change in their social activities. Their relations with parents, relatives, friends, and life partners were also affected. These two major changes generated different emotional states. Thus, we can say that the sudden change in traditional activities is mirrored in the social and the subsequent emotional changes. The students’ sense of belonging to the university community may have been lost, bearing in mind that they could only have a direct connection to the learning environment or to social activities in an online format. The lack of social contact had a particular impact, and the emotional effects were catalyzed by the news they watched and read, the lack of a regular routine, the lack of communication, the decrease in income, the lack of social interaction, etc.

The negative effects arose not as a result of a low degree of adaptability of young people to the new pandemic situation, but rather in response to the multitude of events that occurred in a relatively short time. In Romania, a state of emergency was declared in March 2020, and in May 2020, the toughest restrictive measures were implemented. Thus, all the social changes mentioned occurred within a relatively short period, consequently provoking significant effects.

Our results show inequalities between students, with students living in rural areas being disadvantaged, in particular as regards access to online educational resources and keeping in touch with the school and colleagues. This is because the resources and logistics needed to gain online access were not sufficient. Therefore, the gradual increase in restrictive measures was directly proportional to the decrease in access to education for students who belong to vulnerable groups, i.e., students without financial resources, without access to the internet, students from disadvantaged areas, from families with problems, or living in inadequate dwelling places [28]. Moreover, the lack of experience in regard to online teaching and learning induced uncertainty, a sense of insufficient knowledge, and the emergence of cognitive gaps [28].

UNESCO has identified large discrepancies between different parts of the world in terms of access to new forms of education. In South Asia or sub-Saharan Africa, for example, 89% of children do not have access to a computer [29]. A report by Common Sense Media and the Boston Consulting Group showed that before the pandemic, between 15 and 16 million American students out of 50 million lived in a household that did not have access to the Internet, to a digital device, or both [30]. There are also discrepancies if one considers the students’ place of residence: 21% of students are from urban areas, 25% from suburban environments, and 37% from rural areas. Differences were also registered from a racial point of view: 30% of the black population had these gaps, as well as 35% of Native Americans [30]. Therefore, the correlation between the residence environment and the educational environment during the pandemic for Romanian students may correspond to a more general picture, of which the UNESCO Director-General Audrey Azoulay warns: “(…) we now know that teaching and continuous learning cannot be limited to online means. (…) In order to reduce existing inequalities, we must support other alternatives, including the use of community radio and television broadcasts and creativity in all modes of learning. These are solutions that we approach with our partners, the Global Coalition” [3].

In Romania, students’ access to educational resources during the closure of educational institutions influenced the education process at both pre-university and university levels. According to statistics provided by UNESCO, Romania had more than 3.5 million students who were affected by the epidemiological context during the state of emergency. An analysis by IRES showed that 32% of pupils did not have access to a functional device (e.g., laptop, tablet, desktop) to carry out their online activities [31]. Moreover, 12% did not have internet connection or did have it but not powerful enough to allow online teaching activities. According to the data provided by NIS, only 66% of households in Romania had access to a computer with internet connection [32]. Moreover, in rural areas, access was even lower, with only 48% having access to a computer and the internet. Our study shows that 56% of those who did not keep in touch with the professors and the university were from rural areas, suggesting that students confronted similar problems.

This research highlights that students prefer the traditional learning-teaching-assessment format, a result that is similar to other studies in the literature. For example, another research conducted in Romania—with 211 responding professors, 208 students, and 152 parents—showed that most of the participants preferred to return to face-to-face education [33]. A study by Jordan University of Science and Technology showed that out of 730 students, two-thirds would prefer face-to-face exams [34]. The factors that accounted for this preference were the difficulty of studying online, the lack of eloquent explanations in the online mode, and the possible inappropriate behavior during exams.

Our research shows that transferring courses to online platforms and the limitations in the pandemic context affected the students’ relationship with the university, both in terms of knowledge and learning as well as in terms of relation, communication, and freedom of expression. The whole situation was perceived by students as a temporary one, a solution to a crisis, and in no way was it considered as a future method for the functioning of the universities, nor did it represent the “new normal”. Moreover, our research shows that for an important category of students, studying, reading, doing sport, or communicating with those close to them have become coping strategies in critical situations. We can therefore say that although education was visibly affected in the state of emergency, through its various expressions, it has become a genuine resource for young people.

Our results indicate a number of emotional vulnerabilities experienced by young people in the face of critical moments generated by the pandemic. A degeneration of effective communication due to the so-called infodemia also contributed to this [35].

The emergence of mental illness is the unseen part of the iceberg of pandemic consequences among students. In our analysis, states of aggression, intolerance, abandonment as well as states of crying, panic, fear, or despair were identified.

We found that isolation imposed in the context of the COVID-19 pandemic meant for the students a lack of social interaction with colleagues and friends, limited movement, and a lack of cultural activities and routine, which they felt had damaging effects on them. These could easily turn into serious sources of frustration and generate different emotional states, especially in the context of the pandemic [36].

A longitudinal analysis, which began in 2018, investigated social networks and the mental health of students in Switzerland [37]. The authors of the study showed that COVID-19-specific concerns led to online isolation, and that the lack of interaction, emotional support, and physical isolation were associated with negative mental health trajectories. In France, out of 3764 students, 22% perceived a high state of stress as a result of the COVID-19 infection of a person in their social network. This generated poor educational results, increased consumption of alcohol and tobacco, and the use of forms of entertainment to calm themselves [38]. A study conducted in May 2020, in seven countries (China, Ireland, Malaysia, South Korea, Taiwan, the Netherlands, and the US), revealed that students from European countries were much more affected by the COVID-19 pandemic as compared to students from Asian countries [39]. When the habits, routines, and behaviors long practiced by students in a certain social and educational system are suddenly cancelled and changed, uncertainty, frustration, psycho-mental imbalances, and adaptation difficulties arise.

The presence of mental disorders in times of crisis is not new [40]. Our results belong to a more general trend of emotional changes during the pandemic period. Official data and recent studies have recorded two broad categories of people affected by these mental imbalances: the infected population suffered from post-SARS DSM-IV psychiatric disorders, post-traumatic stress disorder, or depression, while the healthy part of the population was affected by anxiety, depression, lack of concentration, irritability, etc. [41]. This fact has been noted since 2019, in a study conducted in China [42]. In March 2020, research on a sample of 1116 citizens in the southern Iranian city of Rafsanjan found that 35% of the participants had mental health problems, anxiety, sleep disorders, and psychosocial imbalances [43].

Recent research has shown that current measures to control the spread of the virus (physical distancing and quarantine) can trigger serious imbalances in mental health [44,45]. The study conducted by Rajkumar recorded a whole casework that described a worrying phenomenon: the onset of anxiety and depression as well as suicide attempts among a large segment of the population. The study conducted by Tang Fang using the Depression Scale (CES-D-20) and the Goldberg Scale (GAD-7) on the degree of anxiety and depression showed that the two imbalances in the mental state were identified in 26.47% (depression) and 70.78% of the respondents [46]. The situation is similar in Romania. The IRES research, as of February 2021, shows that one in five Romanians felt more alone during this period, with the need for contact with other people being more stringent among women, young people, and people over 50 years old. It is also worth noting that “1 in 10 Romanians has gone through anxiety states in the last 7 days almost daily, 13% felt sad or hopeless, 14% could not control their worries, and 14% felt they had no interest or pleasure in doing anything” [18].

Socially, relationships with friends and colleagues were affected by strong negative circumstances: lack of social interaction, lack of face-to-face communication, or lack of joint activities. It should be noted that throughout the period of the pandemic, these elements of direct mutual relationships were highlighted in different analyses [37].

Our results show that during the isolation period, the relationships of the respondents with their neighbors were the least affected, followed by relations with siblings, parents, and relatives. In practice, the pandemic and the state-of-emergency measures have reactivated the resources of the “warm community”, in Georges Ballandier’s terms. In these communities, which are marked by deep organic solidarity, the students could reset their lifestyle and priorities. The isolation scenario brought to the fore social networks characterized by solid, time-checked links with a strong affective component. Parents, spouses, lovers, relatives, true friends, and neighbors represent in the equation of our lives a “significant other”, built through a direct, face-to-face relationship [47]. Thus, planning the “stay-at-home situation” during the isolation period meant a reset of lifestyle and relationships with others. Direct human relationships were replaced by those in the virtual environment, keeping only close relationships with family, relatives, and close friends. The social circle of living interactions significantly shrank, and the former social life faded into a virtual universe or became a nostalgic memory.

Our study reveals that the absence of direct interactions has led to the degradation of the relationships of the participants with their friends and colleagues. This result demonstrates that social networks and new means of communication fail to replace living, authentic human relationships. At best, new technologies maintain the illusion of a relationship, becoming a kind of simulacrum of reality that proves its ephemerality in times of crisis [48]. Likewise, online education may have the same effects. The fact that it is perceived as a crash solution by students demonstrates that technology, however sophisticated, cannot replace living interaction.

Social cohesion and human relations have suffered throughout this period. Studies show that in different countries, social cohesion was much lower in June 2020 than in other pre-pandemic intervals [49].

Placed within a broader context of national and international research, our results are consistent with a general picture of the crisis:-The state of emergency has had visible effects on students’ social networks; refuge was sought in different communication spaces and direct interaction waned;-The emergency state also had an impact on the emotional sphere. The emotional states experienced by the students during this period heralded a series of experiences outside of the comfort zone with which they were accustomed. They produced a number of imbalances that were felt at a moderate and medium level of intensity;-Constraints and limitations of the state of emergency have led to a reconfiguration of lifestyle, a change of priorities, and a reactivation of religiosity;-Significant family and social networks have maintained direct relationships. They have been an oasis of normality and real sources of emotional support and help in any situation;-Intellectual activities (study, reading) and hobbies (sport, music) have made the period of maximum isolation more bearable and easier.

A new direction of study starting from the results of our research could be an analysis of university drop-out as well as the educational situation of universities in 2021 as a consequence of the abovementioned pandemic effects.

## 5. Strengths and Limitations

The strength of our research is that data were collected during the state of emergency, which allowed for a better understanding of the analyzed phenomenon. At the same time, we collected first-hand information, untainted by the passage of time and other experiences. Furthermore, this was a nationwide study, which included a large number of participants (N = 1.013).

A limitation in the sampling process results from the online distribution of the questionnaire. However, this approach proved to be the most useful data collection tool at a time when the restrictions were at a maximum, in accordance with Decree no. 195 of 16 March 2020 issued by the Romanian Government.

Given that the questionnaire was distributed online, the sample was one of convenience [50], which limited the statistical representativeness of the results. However, the focus of our research was on understanding and not necessarily on generalization, on the basis of representativeness. This limitation is offset by the large number of respondents and the fact that the research was carried out at the very peak moment of restrictions against the backdrop of the pandemic.

## 6. Conclusions

Our research provides important insights into the educational, emotional, and social impact of the COVID-19 emergency state on Romanian university students.

We found that students accepted online education only as a form of compromise in relation to the epidemiological situation. They were affected by the transition to the online format, owing to little contact with the university and other students, and the lack of a regular routine.

On an emotional level, we found higher values in relation to the feelings of loneliness, panic, fear, aggressiveness, and intolerance due to the lack of cultural activities, the lack of the usual routine, and the limitation of communication and movement.

Socializing with friends and fellow students was also severely affected. The closure of universities led to their return home to parents, who were elements of support for the students in the context of the pandemic.

The results of this study established a number of trends in the self- assessment of students’ personal experiences during the state of emergency. The state of emergency during the pandemic period will remain a special experience in the students’ lives. Its dramatism was tempered well by an effective support mechanism, provided by solid social ties, intellectual formation, and a certain religiosity. This has produced good resilience among young people, but also in the communities they belong to.

The period of confinement can also be seen as an initiatory experience from which young people emerged stronger and more mature. In the long term, this moment could cause changes in the meanings of social relations, choices (professional, living space, consumption, travel, leisure practices), and—why not?—the metaphysical dimension of existence. It remains to be seen whether the COVID-19 pandemic will be remembered as a life lesson in the collective mind of students and beyond.

## Figures and Tables

**Figure 1 ijerph-19-03990-f001:**
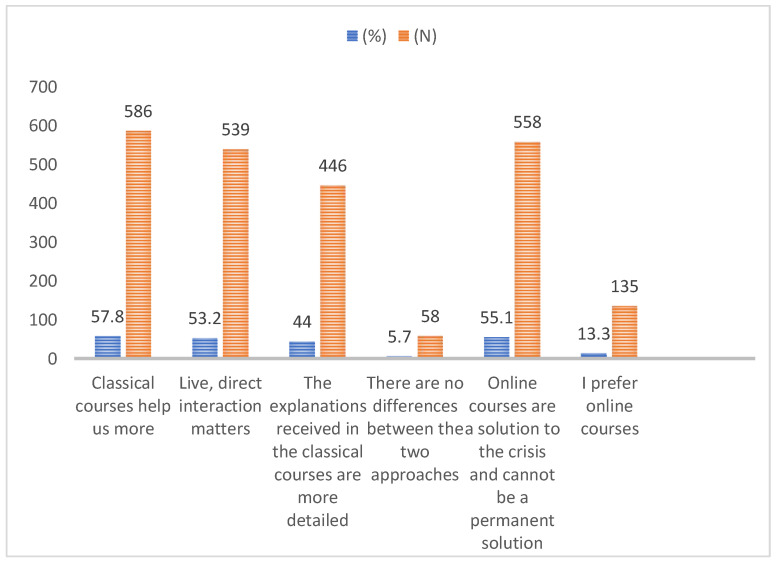
Opinions on the format of the courses.

**Figure 2 ijerph-19-03990-f002:**
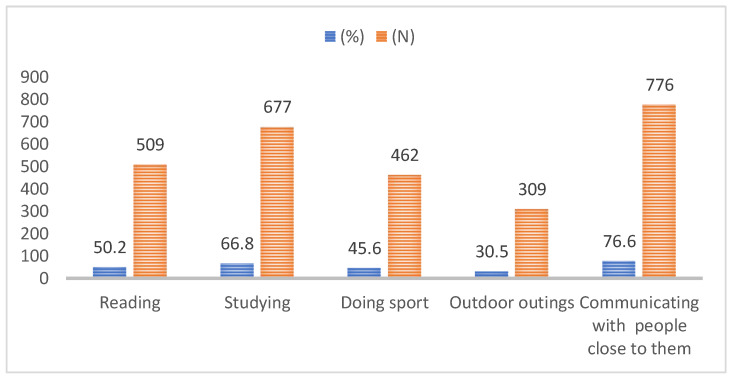
Activities that students did not give up during the state of emergency.

**Figure 3 ijerph-19-03990-f003:**
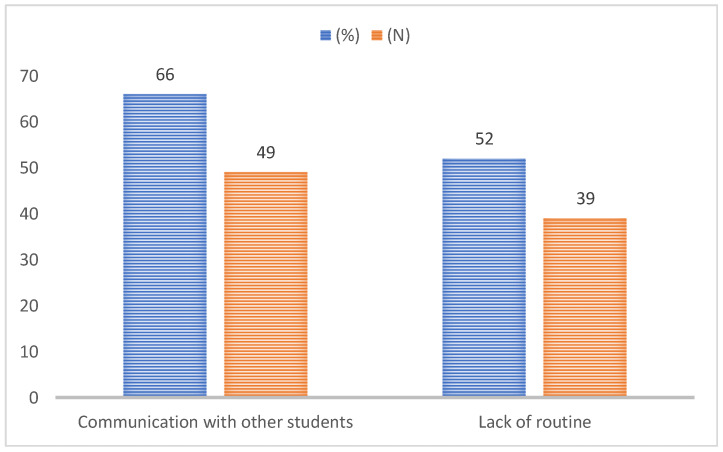
Consequences of the loss of contact with universities.

**Table 1 ijerph-19-03990-t001:** Analysis dimensions and indicators.

Dimensions	Indicators
**Educational Dimension**	- Relationship with teachers; - Teaching, learning, assessment; - Access to educational resources.
**Emotional Dimension**	- Emotional state during the period of isolation; - Prioritizing personal goals; - Effects of isolation on routine activities
**Social Dimension**	- Relationship with people in personal social networks; - Level of socialization

**Table 2 ijerph-19-03990-t002:** The structure of the study group.

Variables	Levels	N	(%)
Age	18–23 23–25	750 162	74.03% 15.9%
	25–30 Over 30	77 24	7.6% 2.36%
Field of study	Science	595	58.7%
	Socio-humanities	418	41.3%
University program	Bachelor’s	842	83.1%
	Master’s	140	13.8%
	Doctoral	31	3.1%
Living environment	Urban	627	61.9%
	Rural	386	38.1%

**Table 3 ijerph-19-03990-t003:** The situation of maintaining communication with the school and teachers.

Residence Environment		Valid	Frequency	Percent	Valid Percent	Cumulative Percent
Rural		No	42	6.7	6.7	6.7
		Yes	585	93.3	93.3	100.0
		Total	627	100.0	100.0	
Urban		No	33	8.5	8.5	8.5
		Yes	353	91.5	91.5	100.0
		Total	386	100.0	100.0	

**Table 4 ijerph-19-03990-t004:** Social and cultural factors that influenced the students’ emotional status during the lockdown.

Factors	Mean
Loneliness	2.39
News	2.34
The ones I have been with	1.84
Lack of regular routine	2.82
Lack of perspective	2.52
Lack of communication	2.61
Lack of cultural activities	2.86
Decrease/lack of income	2.46
Lack of physical interaction with colleagues	3.30
Lack of movement	3.06

**Table 5 ijerph-19-03990-t005:** Emotional reactions during the state of emergency.

Reactions	Mean (1–5)
Panic	1.83
Fear	2.06
Boredom	3.11
Peevishness	2.21
Intolerance	1.95
Aggressiveness	1.79
Despair	2.01
Crying	1.86
Abandonment	1.74

**Table 6 ijerph-19-03990-t006:** Effects of isolation on students’ social relations.

Relationships	Mean	Variance
With parents	1.53	0.747
With siblings	1.44	0.594
With relatives	1.53	0.623
With neighbors	1.41	0.527
With friends	2.01	1.052
With colleagues	2.05	1.089

## Data Availability

The data presented in this study are available by request from the corresponding author.

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
