# Peer review of "Educational, Emotional, and Social Impact of the Emergency State of COVID-19 on Romanian University Students"

_ijerph, 2022, doi:10.3390/ijerph19073990_

Round 1
Reviewer 1 Report
Dear authors:
I consider the changes made to be appropriate. Before the article is published, there are typographical errors that need to be corrected (paragraph spacing, punctuation marks, etc.).
Best regards.
Reviewer 2 Report
Dear authors, thank you for your efforts. Several of the corrections requested were not made (e.g. first few lines of the abstract and the very long sentence on page 3). Further to this, several unrequested changes were made to the structuring of the article (e.g. introduction and results). Could you kindly explain why this is the case?

Reviewer 3 Report
Thank you for your revision. However, there are still some errors in this manuscript. Therefore, I suggest major revision. Please pay attention to the following comments:
- After revision, the structure of the paper getting worst. For example, line 39, only a paragraph with a line? This happened in the whole manuscript and seems the authors did not pay attention to the structure of the manuscript. It is very messy and far from a common scientific paper. The authors can combine most of the paragraphs as they are talking about the same topic
- Why there is not any reply to reviewers file which provides response to each comment?
- In introduction, expand the emphasis on the subject matter by focusing on recent articles in the International Journal of Environmental Research and Public Health.
- Line 132-145 please revise this section.
- As most of your collected data from individuals are subjective, how we can trust your findings?
- To give more information for readers and enhance the quality of paper, it is recommended that the results represent as a column figure.
- Please modify your references format. Seems some of your references are not in English. So clarify them by adding a phrase at the end of them (e.g. in Romanian)

Round 2
Reviewer 3 Report
Thank you for your effort. The authors addressed most of my concern, so the publication at this stage is acceptable.
This manuscript is a resubmission of an earlier submission. The following is a list of the peer review reports and author responses from that submission.
Round 1
Reviewer 1 Report
Dear authors:
The article you submit is of interesting, and provokes reflections of importance for the educational field.I suggest some modifications:
- In the introduction of the work, general data of little interest for the study are included. On the contrary, the dimensions that are the conceptual basis of the empirical study could be explained in more detail, in relation to the importance of the emotional, social and educational aspects of education itself.
-The methodological description could be improved. In this respect, what research method is being applied? There is also a need for further explanation of the instrument and the nature of the variables. Otherwise, the analysis is not sufficiently understood.
-The tables include information that is not necessary, while there is other data that is not necessary, so that they are sometimes unclear.
Author Response
"Please see the attachment."

Reviewer 2 Report
Thank you to the author/s for this interesting article. I think that a lot of work has gone into it.
However, it needs further work.
- Firstly, specific research questions to explain what was guiding the research are needed.
- Secondly, an expanded and focused literature review is needed with a greater emphasis on university students. Perhaps this could follow the introduction.
- Thirdly, the connections between some statements, points etc. need to be made explicit.
- Finally, the overall language and grammar require work and editing.
Best wishes.

Author Response
"Please see the attachment."

Reviewer 3 Report
The manuscript title: " The manuscript title: " Educational, Emotional and Social Impact of The Emergency State Of Covid-19 On The Romanian Students" seems interesting work, however, there are several errors in this work. Therefore, I would like to recommend this manuscript for "Major Revision".
- Based on the manuscript content, it’s necessary to alter the title to fit the content (e.g. add university to the title).
- The abstract needs to be rewritten to briefly state the purpose of the research and to draw meaningful conclusions based on the results obtained.
- The number of key words is exceeded the routine (maximum 5).
- In introduction, expand the emphasis on the subject matter by focusing on recent articles in the International Journal of Environmental Research and Public Health. Authors must modify introduction by stating the following points "Problems, Possible solution, Disadvantages of these solutions, Author's idea, Advantages of authors idea, etc." Introduction in overview form is not recommended. Also, the authors talked about the effect of COVID-19 on environment. So, please use the following references for the modification of the introduction. https://doi.org/10.1016/j.ceramint.2021.08.042 https://doi.org/10.1038/s41598-021-83166-4
https://doi.org/10.1007/s10278-019-00209-z - To give more information for readers and enhance the quality of paper, it is recommended that the results represent as a column figure.
- Rewrite the conclusions. The conclusion should be enriched by the important results of the whole study.
- There are several grammatical errors in the manuscript. So, the language must be polished throughout the manuscript before publishing.
- The questionnaire should be included in the manuscript as supplementary section.
- Line 154 – check the number of participants
- How many university centers were involved in this study?
- What method you used to analyze your data?
- Which software was used?
- As most of your collected data from individuals are subjective, how we can trust your findings and your conclusions?
- Please modify your references format. Seems some of your references are not in English. So clarify them.

Author Response
"Please see the attachment."

Round 2
Reviewer 2 Report
Thank you to the authors for making a great improvement on the first version of the article. This version was better organized and easier to follow. Getting the article proofread was a good idea. However, there are still some minor edits needed. Kindly attend to these.

Reviewer 3 Report
After careful consideration of the revision, I found that the concerns which made by the reviewers have been not well addressed. Also the comments and suggestions did not cited by the authors. Also, the authors ignored some previous revisions comments and questions. Considering that, this manuscript does not merit and not suitable to be published in IJERPH. Therefore, I reject it.